# A Socio-Cognitive Review of Healthy Eating Programs in Australian Indigenous Communities

**DOI:** 10.3390/ijerph19159314

**Published:** 2022-07-29

**Authors:** Jessica Harris, Julia Carins, Joy Parkinson, Kerry Bodle

**Affiliations:** 1Social Marketing at Griffith, Griffith Business School, Griffith University, Nathan, QLD 4111, Australia; j.carins@griffith.edu.au (J.C.); j.parkinson@griffith.edu.au (J.P.); 2Department of Accounting and Finance, Griffith University, Gold Coast, QLD 4111, Australia; k.bodle@griffith.edu.au

**Keywords:** aboriginal, indigenous Australians, nutrition, nutrition education, social cognitive theory

## Abstract

Purpose: This paper aims to understand the challenges to healthy eating for Indigenous Australians using a Social Cognitive Theory lens. Understanding the environmental, cognitive, and behavioural barriers to healthy eating for Indigenous populations in Australia will help identify current gaps and highlight future actions needed in this area to close the gap for Indigenous Australians. Study design: Narrative review of interventions of healthy eating programs in Australian Indigenous communities sourced using a systematic search protocol to understand the environmental, cognitive, and behavioural barriers to healthy eating among Indigenous Australians and to identify gaps and future actions needed to address this from 2010–2020. Results: The search produced 486 records, after duplicates were removed and the inclusion and exclusion process were utilised, seven interventions were retained in nine studies. The seven interventions had multiple study designs, from randomised control trials to case studies. Conclusions: Further work needs to explore the long-term feasibility of providing fruit and vegetable discounts and the impact of remoteness for the delivery of healthy food. Dietary interventions need to be clearly described, and fidelity and process of the design and implementation process to help with replication of work.

## 1. Introduction

The health status of Indigenous Australians tends to be much poorer than the majority of the non-Indigenous population [1,2,3,4]. Since colonisation, the introduction of Western diets, and the disruption of their cultural and traditional existences, there has been an increased rate of poorer health statistics [5]. The estimated life expectancy of Indigenous Australians (76 years for males and 75 years for females) is markedly less than the general population average (80 for males and 83 for females) in Australia [6]. This reduced life expectancy results from many issues contributing to high mortality rates, including susceptibility to preventable diseases, inadequate diet, poverty and limited health resources. Indigenous people suffer from a higher prevalence of non-communicable diseases (diabetes, high blood pressure, obesity and respiratory diseases) than the general population, associated with poor nutrition and sedentary lifestyles habits [7,8,9]. As these disparities continue to rise, research suggests interventions developed to combat these issues cannot be comparable to those of other non-Indigenous communities [1]. Whilst the experience of Indigenous communities located in regional and remote areas may not apply to all Indigenous people—there remains a need to consider how these disparities, especially for regional and remote communities, can be reduced. In recognition of these disparities, the Australian Government intends to ‘close this gap’ within one generation [10,11]. The government-set agenda is problematic, as it positions Indigenous people as dysfunctional and in deficit, requiring remedial action from a capable authority, to ‘normalise’ their behaviour [12]. Despite this precarious position, but considering the critical state of poor health in Indigenous Australians, there is a need to understand the challenges to improving health behaviours, including healthy eating, by examining Indigenous Australians’ health and wellbeing programs [11].

Most health and wellbeing programs aim to establish healthy behaviours that will prevent health problems from arising later. Studies show the influence of behavioural patterns and habits developed in early years continues through the life course [13,14,15]. This may be associated with social determinants of health and increase the amount of early onset of preventable diseases such as (diabetes) and cardiovascular conditions. These conditions can arise from poor nutrition and sedentary lifestyles [7,16,17]. Current research supports the relationship between poor dietary habits in adolescence and obesity, indicating that adolescence may be the prime time to intervene with healthy lifestyle habits [17,18,19,20,21,22]. Poor nutrition can come from a multitude of complex influences from individual, environmental and behavioural factors [23]. Studies suggest that healthy dietary habits can help a person avoid succumbing to the temptation of unhealthy foods [24] and can be maintained by an environment that supports that pattern of behaviour [23]. There is evidence to suggest that obesity can emerge from a young age [25,26,27], with others suggesting the period when adolescents shift into adulthood is thought to be a critical time when unhealthy eating and behaviours may become more common [21], and this is a critical time to encourage the development of healthy eating patterns due to the development of independence and increases in self-efficiency of emerging adults [28]. Taken together heathy eating approaches should be taught from a young age and continued through the life to ensure emerging adults have the correct support, knowledge and understanding of healthy food choices.

Indigenous communities can be defined in the following way “Indigenous communities, peoples and nations are those which, having a historical continuity with pre-invasion and pre-colonial societies that developed on their territories, consider themselves distinct from other sectors of the societies now prevailing in those territories, or parts of them. They form at present non-dominant sectors of society and are determined to preserve, develop and transmit to future generations their ancestral territories, and their ethnic identity, as the basis of their continued existence as peoples, in accordance with their own cultural patterns, social institutions and legal systems [29]”.

Health promotion is an important part of facilitating and advancing Indigenous communities as it enables longer and productive lives and reduces non-communicable diseases. Using health promotion through cultural and traditional practices can assist in creating better living conditions and a longer lifespan. Adopting healthy practices early and maintaining them throughout the life course can prevent the emergence of health problems, and disease, which in turn reduces the need for demanding or expensive medical care. A few health promotion campaigns have been conducted for Indigenous youth, which has provided intervention protocols and outcomes. However, the review of nutrition education programs by Kagie, Lin [16] advised many do not describe the implementation of the nutrition components of the programs, and they do not have a high success rate without follow-up procedures which limits program sustainability past the program duration. Additionally, the influence of parents, social groups and community in the adoption of healthy practices can be very important [30,31]. Understanding the socio-economic benefits that health promotion and interventional support can create with and for Indigenous populations in the context of obesity and other dietary-related diseases is important [30].

Healthy eating programs currently ignore historical Indigenous cultural backgrounds and their way of eating and generally only account for western ways of healthy eating rather than reconnecting with Indigenous worldviews as a catalyst to improved health [32]. Many western foods are highly nutrient-dense compared to Indigenous diets pre colonisation [33]. As a result, healthy eating programs are not designed with cultural knowledge or community involvement [5]. Four previous reviews that focussed on health promotion for Indigenous people found the main gaps in the evidence were implementing sustainable nutrition programs, comprehensibly describing the implementation process, lack of evaluation methods, lack of culturally appropriate material within interventions to include traditional food sources, not addressing health services within the environment and little community involvement in program design [7,11,16,33]. This review takes on a different approach to systematically look at healthy eating interventions from the past ten years to strengthen the case from these past reviews that more work is needed to be done in Australian Indigenous areas to increase healthy eating using a socio-cognitive approach.

Theory helps to provide a basis for designing, implementing and evaluating complex interventions [34]. Calls have been made for behaviour change programs utilising social marketing tactics to provide a strong theory-based method and a complete account of how theory has been used [35,36]. Understanding the causal relationship between behaviour and intervention effectiveness can be strengthened by rigorously tested design methods that target theory use [34]. Behaviour change programs currently under-report theory use, contain poor theory development and reproducibility, and do not use a comprehensive evaluation of the program [36,37,38,39]. There needs to be deepening and broadening in behaviour change programs, by practitioners and researchers alike in the social marketing field, to include more utilisation of theory. Theory facilitates understanding of behaviour, and the design of interventions and underpins rigorous evaluations to determine program effectiveness. This has been identified as a key area for improvement in behaviour change and social marketing [38]. Social marketing can be defined as: “*the activity, set of institutions, and processes for creating, communicating, delivering, and exchanging offerings that have value for customers, clients, partners, and society at large*” [40]. More broadly social marketing helps enhance social good and change human behaviour by working with the people not for them [41]. Social marketing seeks to influence behaviour in health and wellbeing and create positive lasting change. Importantly, social marketing offers a great deal as an approach for approaching indigenous health promotion because of the emphasis it places on understanding the people at the heart of any issue, and the context in which these people live, as well as the commitment social marketing places on developing change strategies with communities to ensure those strategies are valued by those communities.

This paper draws on Social Cognitive Theory (SCT), a widely used theory in health interventions, social marketing interventions, and notably the most referenced theory in healthy eating behaviour studies [42,43]. SCT describes three domains of influences on behaviour: environmental, cognitive, and behavioural. Importantly, SCT considers the behavioural context to move beyond the individual, which is likely to include important influences within Indigenous communities. The model that was used within this study was Bandura’s reciprocal triad [44,45]. Within this triad there are nine constructs, the individual factors of knowledge, expectations and attitudes; and the behavioural factors of skills, practice and self-efficacy; and the environmental factors of social norms, access and influence. SCT distinguishes the reciprocal determinism of behaviour, where both the person influences their environment and the environment influence the person [41]. Theories such as SCT help practitioners link behaviours and influences together to provide the foundation for understanding effective design and evaluation [46].

This paper aims to understand the impact of healthy eating interventions for Indigenous Australians using a SCT lens. Understanding the environmental, cognitive and behavioural barriers to healthy eating for Indigenous populations in Australia and the effect of program strategies in these three domains on healthy eating will inform the design of future programs. Collating their limitations and gaps will help identify and highlight future actions needed to ‘close the gap’ [10] for Indigenous Australians and provide insights for intervention development in the coming years.

## 2. Materials and Methods

### 2.1. Search Strategy

This study performed a systematic literature search to find publications of healthy eating programs conducted with Indigenous Australians and then reviewed these programs to examine how they have tried to improve healthy eating, the effectiveness of the programs, and the lessons learned. The search strategy used four databases, Medline, Psych INFO, Web of Science and ProQuest. The search protocol used the following key terms, grouped into three sets of terms: “healthy + eating” OR “food + consumption” OR “nutrition” OR “diet” OR fruit” OR “vegetable” AND “Aboriginal*” OR “Aborigine*” OR “first + people*” OR “ATSI” AND “intervention*” OR “Randomi#ed + Controlled + Trial” OR “evaluation” OR “trial” OR “campaign*” OR “program*” OR “experiment” OR “study” OR “studies” OR “initiative”. Searches were restricted to the peer-reviewed scholarly literature to provide a level of assurance of the quality of the research, validity of the data, and the veracity of the study conclusions. Results were limited to those published on or after the year 2010. Searches were restricted to the peer-reviewed scholarly literature to provide a level of assurance of the quality of the research, validity of the data, and the veracity of the study conclusions. Given the Shadow Report (2010) aimed to initiate significant and coordinated action to ‘close the gap’ [10] for Indigenous Australians, reviewing studies from 2010 onwards would provide a current perspective of research on this topic and would be reflective of any action taken. A PRISMA flow diagram was used to document the search process and capture decisions made during the inclusion and exclusion process.

### 2.2. Inclusion and Exclusion Criteria

An exclusion process was used to analyse and exclude any articles not related to the topic. All database results were combined and ordered, and duplicates were removed. After the initial ordering of papers, each title and abstract were screened. The following exclusion criteria were applied during screening: (i) papers that were disease-related or not focussed on nutrition behaviours (ii) papers not in English, and (iii) papers where the focal population was not Indigenous Australians. Papers that were disease related were excluded to ensure the review focussed on preventive health, rather than medical intervention. The full articles for any remaining records were examined, and articles were only included in the review if they focussed on Australian Indigenous people; involved a program or intervention; the intervention was not disease focussed; and the intervention aimed to increase healthy eating (including but not limited to fruit and/or vegetable intake). Two researchers conducted this process, the lead author who identifies as an Indigenous person. Backward and forward-searching were conducted to identify other papers associated with the identified studies that met the inclusion criteria. Each record during the exclusion and inclusion process was assessed independently, and all differences were resolved through discussion between two researchers to form the final group of articles to review.

### 2.3. Review Approach

A data extraction template was created to systematically capture the following data from each of the articles in the final group of studies: details of the study (author, year of publication), program name, participant details (sample size, characteristics), details of the intervention and strategies, research/evaluation design, duration of the study, a theory used (if any), constructs or measurements, study findings and gaps, limitations or lessons learned. Two researchers reviewed all articles presented in the summary table to ensure accuracy and reliability.

### 2.4. Assessment of Study Designs

The National Health and Medical Research Council (NHMRC) [47] framework was used to grade the study evaluation designs from I (highest) to IV (lowest), to assess the level of evidence each evaluation could contribute to the evidence base.

### 2.5. Assessment of Studies Using Social Cognitive Theory

As many SCT constructs are targeted and measured independently of the use of SCT, an assessment of each of the studies was completed to analyse what constructs of SCT were used throughout the studies even if the authors did not explicitly state that they had used SCT. Table 1 provides a full breakdown of the constructs used within SCT that were not theoretically mapped within each study, the study, sample and intervention type.

## 3. Results

The search produced 486 records, after duplicates were removed and the inclusion and exclusion process were applied, seven interventions in nine studies were included. This process is shown in the PRISMA flowchart below in Figure 1.

There was heterogeneity between the study design protocols used to evaluate the healthy eating programs in Indigenous communities. There was one randomised control study [49], one quantitative case study [48], two mixed methods studies [51,52], one cohort pre and post surveys [55], one case control study [53], and one qualitative case study [54]. The studies employed different outcome measures, and due to this variation, and the small sample of studies, a meta-analysis was not possible. Therefore, a narrative review was undertaken to learn as much as possible from the studies. The full table can be found in the Appendix A.

### 3.1. Assessment of Study Designs

Under the NHMRC assessment framework, one study was rated as level II, three studies level III-2, one study III-3 and two studies level IV. Most interventions (*n* = 3) were comparative studies with concurrent controls but had minimal strategies to prevent bias and reported weak study designs that were not described in full. Many articles did not provide data collection methods to the full extent that would enable them to be reproduced, nor did they provide evidence that the data collection tools were reliable or valid. In future research, designs need to report program design, methods and results for reproducibility fully. Within the review there were four of the nine studies that reported some use of best practices [48,49,51,52]. These papers discussed best practices of partnership and collaborating with the community, consulting them for their input into the interventions and hiring Indigenous people into the programs. However, it was evident that more could be done to incorporate best practices within these studies, and details of these practices need to be reported in full to enable assessment with tools such as the CREATE QAT [48].

### 3.2. Behavioural Focus

The studies varied in the behaviour they sought to change. There were (*n* = 5) studies that sought to change healthy eating [48,49,52,54,55], (*n* = 1) study that sought to increase fruit and vegetable intake [51] and (*n* = 1) that looked at increasing breakfast consumption [53]. The major strategies programs used to change behaviour included sales (including promotion and pricing strategies), education, environmental changes and community engagement. The utilisation of sales, promotion and pricing was one of the most used strategies within interventions; due to the high cost of fresh, healthy food in remote Indigenous communities pricing strategies were often seen to increase healthy eating [48,49,51,52].

Furthermore, education was an intervention strategy that was used with most of the studies. The process of educating communities to understand information on food packaging, long-life foods and the knowledge to incorporate both traditional foods and store-bought foods into their diet were the main aspects of the interventions. These intervention strategies were delivered through classes [49,53], yarning circles [54], and engagement with the community [48,54,55]. Additionally, modifying the environment was a further strategy utilised [48,51,52,53,55]. Increasing the availability of food through the food system and supporting the feasibility of having fresh fruit and vegetables regularly, overhauling the infrastructure for keeping food and facilitating with store owners to achieve these outcomes. All seven studies sought to use more than one intervention strategy, with most incorporating educational and environmental strategies to increase healthy eating.

### 3.3. Theoretical Focus

Within the seven studies, reporting of any theory was rare. There were five studies that did not report the use of theory during design and/or implementation [48,51,52,53,55] and two that discussed the use of theory to be able to design for reproducibility [54,57]. One study utilised Social Cognitive Theory [49], and one was guided by the Making two Worlds Work (MTWW) framework [54]. However, in the studies many intervention strategies under the domain of SCT were not reported. This is problematic for practitioners who want to replicate these studies. Therefore, the following the section describes the strategies and measures used under each domain of SCT (behavioural, cognitive and environmental) that should have been addressed with the studies. From Figure 2, knowledge and access to food were the two mostly used constructs within the SCT framework. With self-efficacy and practice were also commonly used. Figure 2 provides a breakdown of the studies that used the constructs and which ones were ignored which is numbered next to each construct on the number of times being used. The behavioural triad was most used with each construct being used and the Cognitive triad the least used with only knowledge used four times.

Behavioural strategies included practicing new skills, developing cooking skills and facilitating lifestyle changes through engagement practices in educational classes. Cognitive aspects included building a knowledge base and confidence in nutrition knowledge and education and understanding labels on store-bought food [49,53,54,55]. Environmental strategies included changing the environment, delimiting the risk of access to food, social settings and understanding the important factors of social interactions and support [48,51,52,53,55]. Utilising price discounts on fruit and vegetable intake to decrease this barrier was also seen in [48,51], providing evidence that environmental factors play a big part in enabling Indigenous communities to access fresh, healthy food. Furthermore, [52] measured intervention effectiveness through feedback channels, recall, and barriers faced in the environment.

### 3.4. Challenges and Limitations within the Reviewed Studies

There were multiple gaps and limitations described in each of the studies as they attempted to increase healthy eating of Indigenous people. The environment was the most significant barrier and was discussed in many studies. The limited storage for fresh, healthy foods had a considerable impact given remote Indigenous communities do not receive daily fresh fruit and vegetable [53,58]. These barriers to obtaining healthy fresh food are primarily a result of the remoteness of these places. Furthermore, the social and environmental barriers with program effectiveness for initiatives that encourage community engagement showed greater empowerment when run by the Indigenous community than those that used external stakeholders [53,58]. A need to strengthen and encourage community gardens and kitchens to link the community involvement in healthy eating patterns was noted [58]. Limited access to nutritious food was also observed [59].

Additionally, challenges were experienced with promotional efforts of an increased price discount to encourage fruit and vegetable intake. Many people either did not know about the discount due to poor marketing material, or the 10% discount was not enough to motivate uptake of buying fruit and vegetables [51]. There was a call for better communication between store owners and the program designers to ensure promotional discounts were applied correctly to maximise the effectiveness of program activities [51]. This highlights the importance of program design and promotion and the need to monitor these strategies for a more extended period with greater communication between stakeholders [52].

Furthermore, there is a need for cognitive strategies (such as nutrition education) to encourage Indigenous people to become familiar with store-bought foods and to demonstrate how to integrate these with traditional food sources in way that supports consumption of both traditional and western foods to support a healthy diet. Indigenous people have expressed a desire for this knowledge, which may be addressed in future program designs [54].

Behavioural strategies mentioned to help provide better healthy eating programs were needed to understand and develop in future programs by utilising cooking skills and further practice in cooking with both traditional and store-bought foods [54]. This correlates with the need to facilitate additional cognitive aspects of building knowledge and nutrition [60].

Beyond the lessons that fall into the three domains of SCT, other teachings were noted in the studies. When non-Indigenous people managed studies, the studies did not have the same level of uptake found when Indigenous people were involved in the co-create and build and delivery of the intervention. One study experienced difficulties fulfilling their community stakeholder food group with Indigenous representatives [48]. The lack of capacity and poor coordination was another challenge noted by [53]. With limited resources in terms of volunteers to run the programs, and difficulties getting Indigenous people into the program in remote areas, it was difficult to execute programs in the way they were intended [53]. These issues point to the need for community-driven not government-driven initiatives, which increase partnership within Indigenous communities and promote trust by the community. This new approach to ‘closing of the gap’ encourages a more authentic way forward that values self-determination and true partnership to create equality to Australian Indigenous people through a collective journey [61]. Taken together, these gaps, limitations, and lessons learned to provide valuable insights for social marketers and Indigenous ways of thinking that are based on the same premise because research needs to be for the people and by the people to provide sustainable interventions.

## 4. Discussion

The purpose of this study was to review previous healthy eating programs conducted with Australian Indigenous people to understand what progress has been made in improving healthy eating for this group. Given the complexity of addressing healthy eating within a group known to have different needs and environmental challenges to the larger Australian population, Social Cognitive Theory (one of the most common theories used in healthy eating studies) was used as a theoretical lens for this review. The review examined program outcomes and the programs’ cognitive, behavioural and environmental elements and collated their limitations and gaps. In doing so, this review helps to identify and highlight future actions needed to ‘close the gap’ [10] for Indigenous Australians and provide insights for future intervention development. The new ‘closing the gap’ agreement was released in 2020 containing 17 targets for education, employment, health and wellbeing, justice, safety, housing, land and waters, and languages. Importantly the new agreement contains strategies to develop an authentic way forward with a true partnership with Indigenous people to deliver services together guided by local community members, councils and Elders. This landmark framework was developed together with the commitment to providing equality to Australian Indigenous people through a collective journey [61]. Whilst it is important to set and commit to targets for improvement, there also needs to be established knowledge of how to successfully intervene to bring about positive change.

Indigenous people have higher mortality rates, higher non-communicable diseases and generally eat less healthy food than the non-Indigenous population. Government policies within the timeframe went for this review (2010–2020) [10,62,63] specifically address and acknowledge the health disparities of Indigenous Australians and the underlying determinants of nutrition in these communities. Despite these policies highlighting the need for action, our systematic search found only a small group of studies that attempted to increase healthy eating of Indigenous people individually or within their communities since the Shadow Report in 2010. The low number of studies may reflect a lack of knowledge of how to successfully intervene within Indigenous communities. Together with the generally low quality of the evaluations, this suggests the need for more research to influence healthy eating in this priority group positively. This finding is consistent with other reviews see [64,65,66] who also noted a predominance of overall weak program quality, poor description, unclear reporting and implementation reliability and validity in the dietary assessment.

While there has been some progress, the findings from this review highlight there are many gaps and limitations when attempting to ‘close the gap’ within the timeframe set. These gaps and limitations included a need to build Indigenous-specific programs with multiple stakeholders, and most importantly, with representatives from Indigenous communities. To develop effective interventions, program design must consider accessible, affordable, socially, and culturally acceptable food. Programs also need to teach both children and the greater community about the need to eat healthy fresh food for a healthy life and reduce the prevalence of chronic disease. Program effectiveness was consistent when community programs were led by Indigenous members [5,66]. Despite recognising the need to involve community members, there were challenges when including the community members in developing and implementing the programs [51,67].

Indigenous people have a strong ideological connection to traditional food [54] and less familiarity, knowledge and understanding of non-traditional foods and long-life food and packaging. Therefore, there is an opportunity to develop programs that align with Indigenous peoples’ connection to traditional food. Exploring strategies to provide culturally sensitive information and approaches to support nutrition advice was an important component in intervention protocols. Programs that considered social groups and integrated nutrition knowledge dissemination through Elders down to children had a positive effect [5,60]. Elders discussed that losing traditional food knowledge of seasonal fruit and vegetable sources and skills of hunting, gathering and cooking, has impacted the current health of Australian Indigenous. Coupled with disempowerment and lack of knowledge regarding western food systems, more work is needed in cooking and nutrition classes for Indigenous communities. Developing structured programs that work with traditional and new knowledge of healthy eating programs would be regarded as a positive intervention strategy [5,54].

There were direct links between related healthy eating behaviours, the social determinants that Indigenous people have in these communities of poverty, and the influence of food access and food security that underpin the food choices some of these people make. For example, in many remote communities, fruit and vegetable prices are high, thus store-bought food and unhealthy eating practices are prevalent [60]. Thus, there is an assumption that targeted price discounts will be necessary to improving health however when applied, there were no discernible effects found. This was likely due to poor design and communication of discount promotion and potentially inadequate size of the discount [52]. These findings are consistent with other reviews aiming to identify studies that improve health [64,66]. Different discount strategies [49,52] and incentives, including a rewards program [51] had high levels of acceptability, demonstrating the potential for this type of program with additional community supports to increase the opportunity for Indigenous communities to eat fresh fruit and vegetables. Limitations were highlighted when pricing strategies were not explicit, and there was limited engagement, collateral and resources provided to ensure program effectiveness [52], but showed promise in other reviews [64] when subsidies were provided. With the cost of food on average 45% more in remote communities than in urban areas, there is an important need to provide more affordable and sustainable fresh food sources [59]. This review has highlighted several limitations regarding interventions and programs developed for remote communities, including limited investment into programs that included promotional and nutritional advice [51].

This review demonstrates the importance of collaborative approaches for achieving improved engagement and uptake of programs seeking to enhance participants’ dietary intake and health. Collaborative approaches accrue knowledge and shared resources between stakeholders, therefore having a better uptake by participants and potential sustainability, with a lower cost than programs that do not take a collaborative approach. For example, to optimise healthy school programs in remote communities which include Elders through providing traditional food knowledge and sustainably practices of maintaining food sources that were related to their totems, hunting procedures, integrated and sustainable programs that address supply, nutrition, education, and social aspects are required to tackle the complex issues surrounding healthy food consumption [53,66,68]. Therefore, there is a need for improved engagement and coordination between external stakeholders to optimise any potential program or intervention aiming to improve health in remote communities and work on long sustainability of these programs. Strategies that can be modified and reproduced in different environments and individuals need to be implemented with a greater emphasis on establishing operational procedures and responsibilities of stakeholders, including guidance from policymakers in terms of cost-effectiveness of nutrition programs and the potential societal benefits these programs can have [48,55,68].

Additionally, perishable items, particularly fresh fruit and vegetables, have a limited shelf life. Infrequent deliveries combined with inadequate cold storage solutions in remote Indigenous communities result in less availability for community members to purchase and consume [53,65]. Furthermore, there is a need to integrate nutrition-related policies and procedures into everyday practices that promote healthy eating for Indigenous communities. This review highlights an important requirement for lobbying and advocating for environmental and structural changes in Indigenous communities, particularly in more regional and remote communities. The food system needs restructuring with increased resources to improve delivery systems, pricing, infrastructure (e.g., cold storage rooms) and better food policies in stores [48,53,60]. Community members have sought to have more control over food delivery, especially store-bought food and more local production of food sources incorporating traditional foods and have highlighted the barriers of high cost and equipment needs [60,65]. Empowering community members to have more control over the food supplies in their communities will provide the opportunity for improving health outcomes in remote communities. Furthermore, providing self-sufficiency and sustainable living opportunities will go some way to facilitating control over all aspects of the food systems [48,65].

The use of theory during the development and evaluation of behavioural interventions is advised to assist with the identification of behavioural influences and the mechanisms for change, to select appropriate behaviour change strategies then, and finally to understand how the intervention creates a change [69]. This review found limited evidence of theory used in the interventions, with little to no theory reported in the articles. Given Social Cognitive Theory is one of the most widely used theories in studies of healthy eating, there was an expectation that this theory may have underpinned some interventions with Indigenous people. However, only one study used SCT. There were no alternative theories used, instead, the majority of remaining studies did not report using theory at all. Some intervention strategies could be classified as targeting one or more of the three theoretical domains of SCT (cognitive, behavioural and/or environmental), giving some insight into effective mechanisms for change. Finally, programs that sought to include cognitive, environmental and behavioural methods into programs appeared to have a high sustainability rate [7,16,33,70]. Moreover, future interventions working within Indigenous communities should also seek to use other culturally sensitive frameworks and concepts that would help strengthen the theoretical base that was lacking in these studies. As noted throughout this review, western theories may not produce that understanding or action most appropriate for Indigenous communities, and researchers should seek to integrate Indigenous worldviews and privilege the voices, experiences and lives of Indigenous people [71].

Theories that should be considered in future studies that align to Indigenous frameworks are Critical Race Theory which offers the viewpoint of race, causes, the magnitudes and inequity, and certain dynamics of cultural implications of power and privilege [72]. The theory explores the shifting role of legal systems for Indigenous people and the movement of critical thought and action [73]. The second theory to take into consideration is Participatory Action Research Theory. This theory identifies the practice of Western and cross-cultural contexts within to perspectives to understand the systematic process through self-reflection and action. It identifies past evidence and builds improvements together to make improvements [74]. With these two theories, future interventions could seek further clarification and understand the most appropriate course of action.

Martin and Mirraboopa [71] take up Rigney’s ideas that Aboriginal researchers need to work ‘‘from the strength and position of being Aboriginal’’ (p. 205) but rejects the notion that this means resisting or opposing Western research frameworks and ideologies. Rather she advocates also working ‘‘alongside and among western worldviews and realities’’ (p. 205). She expands on Rigney’s principles to incorporate recognition of Indigenous worldviews and honouring social values, as well as emphasising socio-historical and political contexts, while still privileging the voices, experiences and lives of Aboriginal people.

The gaps highlighted in this review are consistent with an earlier systematic review that looked at Indigenous programs in Australia with no date exclusion criteria [33]. Individual program effectiveness was less than group programs, providing that social group learning showed overall positive experiences [64]. Secondly, programs encompassing behaviour change strategies with SCT constructs (individual, social and environmental aspects) had greater positive uptake, providing evidence that SCT constructs may be the best way to implement behaviour change programs in Indigenous communities [70]. Interestingly, the review of interventions in Indigenous communities see [33] which included SCT determinants of social (inclusion of collaborative partnerships), environment (changes in the food environment), cognitive (cooking class and increased knowledge) showed improved nutritional intake and had positive effects on the inventions, compared to interventions that did not use these constructs. This review provides evidence that programs incorporating dimensions of SCT can improve program effectiveness through the integration of environmental [48,51,52,53,55], cognitive [49,53,55], and behavioural constructs [54,55].

## 5. Future Directions/Limitations

The limitations of this review include a small pool of studies identified for review and low quality of studies. More research, conducted in a culturally appropriate way, is needed to identify what works to increase healthy eating in Indigenous communities. Ideally, this research needs to be subjected to peer review processes and then disseminated so that a reliable body of evidence can be established to support decision making. The large heterogeneity of program and evaluation design made it difficult to draw definitive conclusions on what programs effectively improved nutritional health in Indigenous communities. Future reviews may need to broaden the search strategy, to include different databases and grey literature to ensure the breadth of effort in this context is considered. This may provide more information on this still emerging area and add to the current picture of action in this area, which may be subject to a lag due to the peer review and editorial publishing process. Studies that discussed disease-related interventions were also excluded (treatment interventions), limiting the review to only healthy eating interventions (preventive interventions). Future reviews may extend to these to determine whether effective strategies working for Indigenous people can be translated from the treatment context to a preventive context. Due to the heterogeneity of the interventions, methods and settings, a meta-analysis was not conducted, limiting the generalisability of the results. Indigenous data sovereignty is problematic within past research, future research should consider integrating principles of data sovereignty into their outcome work and plan for meaningful and respectful decision making when comparing Indigenous and non-Indigenous health outcomes.

There were many programs and interventions (*n* = 184) found in the search to understand the determinants of healthy and unhealthy eating practices in Indigenous communities globally [5,75,76,77,78,79,80,81,82,83,84] in recent years. The results of this study highlight the lack of studies in Australian Indigenous communities. As this is an emerging area, future work needs to be carried out to increase behavioural, environmental and cognitive elements for Indigenous communities. Future interventions need to develop effective interventions that are reliable and easy to reproduce whilst utilising theory-based models to provide fidelity of the work. Program designs must consider barriers of access, scarcity, affordability, knowledge and social and culturally acceptable food. Recommendations for future interventions, including overcoming the barriers of location, socio-economic limitations, poverty, knowledge, the inclusion of all stakeholders within the community and consideration of cultural backgrounds, have been made to ‘close the gap’ [10] for Australian Indigenous healthy eating programs.

## 6. Conclusions

This research attempted to summarise what has worked well and what has not worked well for Australian Indigenous communities to increase healthy eating within remote Australian communities. Whilst there were few healthy eating interventions that specifically encouraged healthy eating, this review has highlighted studies that have increased fruit and vegetables, increased knowledge and participation within these communities. However, the review also highlighted there is a long way to go to develop and implement successful and enduring healthy eating programs in these communities to create the degree of change needed to ‘close the gap’ so that the health of Indigenous people is comparable to that of non-Indigenous people in Australia.

Additionally, the review has uncovered those educational programs are less sustainable when communities are not involved in the design and layout of the programs and helps strengthen work for practitioners in the future. This review has illustrated the need for a co-designed or participatory approach that includes all stakeholders comprising of consumers, health workers, and other community members in the intervention design and evaluation process to develop an effective, culturally appropriate and sustainable intervention in these communities, which aligns with the recent ‘closing the gap’ agreement.

## Figures and Tables

**Figure 1 ijerph-19-09314-f001:**
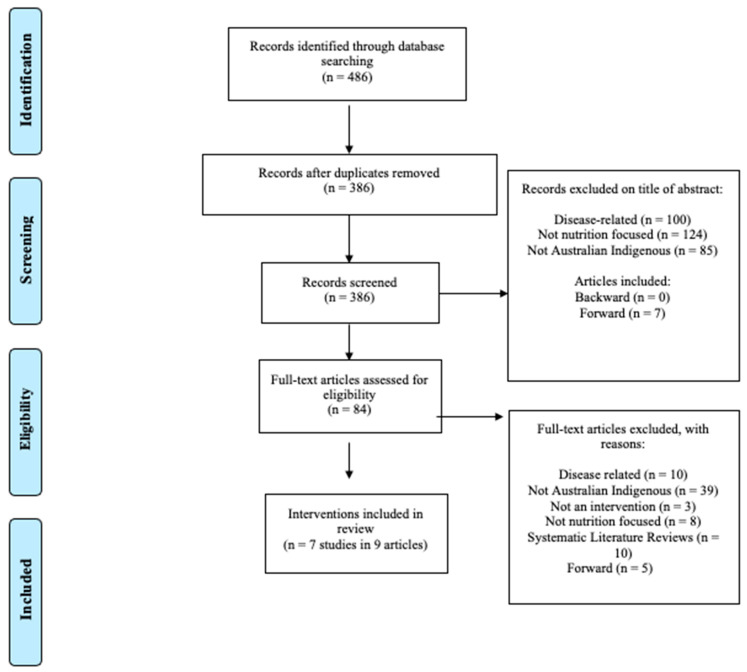
PRISMA 2009 flow diagram.

**Figure 2 ijerph-19-09314-f002:**
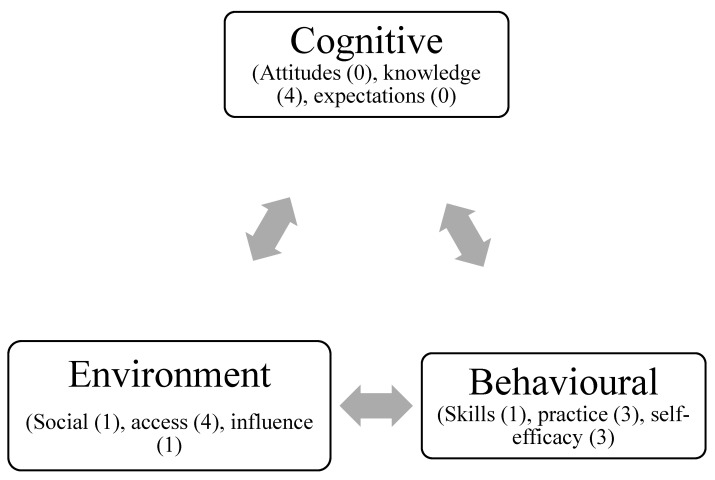
Social Cognitive Theory constructs used within the studies.

**Table 1 ijerph-19-09314-t001:** Summary of healthy eating studies applying SCT.

Study	Sample	Intervention	Behavioural (Individual)			Cognitive (Individual)			Environmental (Social and Structural)			
			Self-Efficacy	Skills	Practice	Knowledge	Expectation	Attitude	Access	Social Norms	Influences	Triads (Constructs)
Good Food Systems Good Food for All Project. [48]	n = 4 aboriginal communities Communities ranged in size (250–2000 residents) Majority (>95%) identifying as Indigenous.	Intervention: Capacity building within the Food System. **Behavioural focus**: healthy eating **Intervention strategies**: Building capacity in leadership, traditional food production, food business practices, food practices in community services and food infrastructure.			√	√			√	√		3 (3)
SHOP@RIC Study Design [49]	n = 20 stores from remote aboriginal communities in the Northern Territory	Intervention: price discount with or without consumer education strategy. **Behavioural focus**: healthy eating **Intervention strategies**: (1) Price discount (20%) on all fresh and frozen fruit/veg, bottled water and artificially sweetened soft drinks. Discounts were promoted in store. (2) Consumer education strategy developed to increase fruit/veg and water intake with monthly themes of health benefits; how much to eat and drink; healthy, quick, and easy meals; supporting family and friends; trying and enjoying new healthy foods; buying more healthy food and making the most of your money				√			√			2 (3)
Substudy (pre-post of the SHOP@RIC study) [50]	n = 20 stores from remote aboriginal communities in the Northern Territory	Intervention: mediators and moderators of SHOP@RIC study **Behavioural focus**: healthy eating **Intervention strategies**: Assessed the impacts of store-based mediators and moderators on consequent diet behaviour. The assessed intake of fruit, vegetable, water and sweetened soft drink. (1) Mediators—perceived affordability and self-efficacy (2) Moderators—barriers and food security.	√						√			2 (3)
Healthy Choice Rewards [51]	n = 1 community in far North Queensland (2500 km from a major city) (1400 residents)	Intervention: Changes in the food system through incentives and sales **Behavioural focus**: fruit and vegetable intake **Intervention strategies**: The Healthy Choice Rewards (HCR) program offered community store customers an incentive of a fruit and vegetable voucher to the value of AUD 10 each time a set minimum amount was spent on fruit and vegetables.							√		√	1 (3)
[52]	n = 6 Indigenous communities (n = 54 participants) across Northern Tertiary and Western Australia	Intervention: price discount using four different strategies **Behavioural focus**: healthy eating **Intervention strategies**: Pricing, infrastructure 4× food and beverage price discount strategies: Reduced price on grocery products. Reduced price on fresh fruit and vegetable Fresh fruit and vegetables at landed cost and Diet soft-drink discount: a reduced mark-up on diet soft drinks.							√			1 (3)
School Breakfast Programs. [53]	n = 2 schools in rural Western Australia.	Intervention: School education program **Behavioural focus**: breakfast eating **Intervention strategies:** An intervention to increase health education, socialinteraction and learning about nutrition and food origins.				√			√	√		2 (3)
Bindjareb Yorgas Health Programme (BYHP) [54]	n = 17 women aged between 18 and 60 years in Western Australia Setting: Regional Bindjareb community in the Nyungar nation of Western Australia	Intervention: cooking and nutrition classes **Behavioural focus**: healthy eating **Intervention strategies** :Recommendations of different food groups to maintain health, development of skills in purchasing healthy food on a budget including reading labels, improve the safe delivery of family foods through safe handling and storage, develop new skills and knowledge in food preparation and cooking and influencing family healthy food choices		√	√	√						2 (3)
FOODcents. Study design [55]	n = 875 (total) n = 706 (non-Indigenous) n = 169 (Indigenous)	Intervention: Adult nutrition education program. **Behavioural focus**: healthy eating **Intervention strategies**: single-session or multi-session courses to increase knowledge and motivation to increase consumption of fruit, vegetables and cereals; decrease consumption of foods high in sugar, fat, and salt; and improve healthy food expenditure	√		√	√						2 (3)
Follow-up evaluation of FOODcents [56]	n = 87 (Indigenous status not known)	Intervention: Adult nutrition education program. **Behavioural focus**: healthy eating **Intervention strategies**: single-session or multi-session courses to increase knowledge and motivation to increase consumption of fruit, vegetables and cereals; decrease consumption of foods high in sugar, fat, and salt; and improve healthy food expenditure	√									1 (3)
Total			3	1	3	4	0	0	4	1	1	

## Data Availability

Not applicable.

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
