# Peer review of "A Socio-Cognitive Review of Healthy Eating Programs in Australian Indigenous Communities"

_ijerph, 2022, doi:10.3390/ijerph19159314_

Round 1

Reviewer 1 Report

The paper reviews the current state of impact of interventions on healthy eating strategies in indigenous communities in Australia, a very interesting review and a thought-through methodology conducted. Very interesting to read! Thanks for compiling this.

The paper gives a good overview about the educative intervention strategies for public health in indigenous communities in Australia.

I only have two major structuring comments that should be tackled all through in order to position this research in reflection to international work content-wise:  

·         How can you locate the SCT within the conducted studies?

I understood that this is the main aim, but this should be elaborated better. The SCT is a very overarching theory that can be applied and synthetized with a lot of other theories. In the column theory and measures you, at the current state, just wrote down whether and which theory was used but not, how you would structure and frame this under the SCT. I would have expected that this review would locate the contents according to this theory, but this should be strengthened and then reflected all over. Even the qualitative content can be allocated there. My suggestion is to add to the table a column for SCT, Reciprocal Determinism, SCT, Behavioral Capability, etc.. and selecte and deduce the meaning derived from the studies already conducted and reflect this in light of the SCT and how this theoretical lens may help to conduct/design further studies in this important field.

·         The table is rather long and crowed; I think this should be a supplementary material. Nevertheless, you could shorten than this and highlight (order) the important points, especially overarching topics

I am very happy to read over this again, rather providing ideas/suggestions for this important piece of work.

Author Response

Thank you reviewer 1. Please see the attached document that answers the questions. 

Reviewer 2 Report

This is an interesting work that aims to analyze healthy eating interventions for Indigenous Australians through a socio-cognitive approach. The article is well-written and follows the PRISMA guidelines for systematic reviews.

Some general comments:

Please include line numbers from the beginning to facilitate review of the article.

Revise writing and typos, for example page 2 last paragraph.

Content is not organized in some parts, mainly in the introduction. Since this review highlights the analysis of health programs using socio-cognitive theory, this approach (theory) should be described in the introduction section, as well as the state-of-the-art or at least indicate why this approach was used. 

Why is it necessary to write “Social marketing can be defined as: “ in bold.

Methods:

In page 23, remover “OR” in the text: “vegetable” AND OR “Aboriginal*” 

Results & discussion:

It would be useful to summarize the main findings and their relationship with the 3 dimensions of SCT using a diagram or any visual representation.

Author Response

Thank you reviewer 2. Please see the attached comments answering the questions. 

Reviewer 3 Report

Abstract: Overall could be more succinct, while adding more details about the search strategy. For example, “This paper explores the environmental, cognitive, and behavioral barriers to healthy eating among Indigenous Australians to identify gaps and future actions needed to address barriers.” This allows for more details so the reader understands what was done.

Introduction: Some minor suggestions-  The sentence starting with “Although well intended…” seems to be an opinion. Introduction should not introduce opinions (suggest to reword to indicate specific ways that this occurs), or provide this type of commentary in the discussion section.

Third paragraph should have a better lead in sentence as the first sentence does not flow well with the content of the rest of the sentence.

In general, because the abstract stated that the review would highlight the recommendations (based on what is found in that review), suggest to keep recommendations for what should be done to after the results of the narrative review have been presented, and not in the introduction section.

The section “Social marketing can be defined as…”  seems out of place. Why are you bringing this up here?

Methods/Results: Were any additional decisions made regarding how to weigh the evidence in the final 9 articles? How was the evidence evaluated using a “social cognitive theory lens? This did not come through in the results section.  

Author Response

Thankyou reviewer 3. Please see the attached document answering the questions.

Reviewer 4 Report

The whole article needs an edit to take out the political statements and to pull the article into a more scientific discussion. A great deal of this paper is more properly placed in social political journals. I didn't find enough recent publications for either the introduction or the discussion. Most research shows that by adolescence obesity is well established and the authors' claim "that adolescence may be the prime time to intervene with healthy lifestyle habits" is not currently the thinking.

The discussion was weak and political and overstated the results. The conclusion " This research has summarised what has worked well and what has not worked well for Australian Indigenous communities to increase healthy eating within remote Australian communities." is not substantiated in the review.

Here are some articles that might help in setting your scene although they do not support starting in adolescence. While they are not Indigenous in focus their results and observations are not based in race but in obesity, the focus of the review and therefore the focus of appropriately targeted interventions for First Nation peoples of Australia.

Ortiz-Marrón, H., Ortiz-Pinto, M.A., Cabañas Pujadas, G. et al. Tracking and risk of abdominal and general obesity in children between 4 and 9 years of age. The Longitudinal Childhood Obesity Study (ELOIN). BMC Pediatr 22, 198 (2022). https://doi.org/10.1186/s12887-022-03266-6

 Brown T, Moore TH, Hooper L, Gao Y, Zayegh A, Ijaz S, Elwenspoek M, Foxen SC, Magee L, O'Malley C, Waters E, Summerbell CD. Interventions for preventing obesity in children. Cochrane Database Syst Rev. 2019 Jul 23;7(7):CD001871. doi: 10.1002/14651858.CD001871.pub4. PMID: 31332776; PMCID: PMC6646867.

 Hennessy, M, Heary, C, Laws, R, et al. The effectiveness of health professional-delivered interventions during the first 1000 days to prevent overweight/obesity in children: A systematic review. Obesity Reviews. 2019; 20: 1691– 1707. https://doi.org/10.1111/obr.12924

 Flynn, A. C., Suleiman, F., Windsor-Aubrey, H., Wolfe, I., O'Keeffe, M., Poston, L., & Dalrymple, K. V. (2022). Preventing and treating childhood overweight and obesity in children up to 5 years old: A systematic review by intervention setting. Maternal & Child Nutrition, e13354. https://doi.org/10.1111/mcn.13354

Author Response

Thank you reviewer 4. Please see the attached document answering the questions. 

Round 2

Reviewer 4 Report

My original comments about the context and tone of the article still stand.  I still think you should take out all the political and poorly substantiated statements about what happened many years ago and what attitudes are today re autonomy in children and how if they were living many years ago they would somehow have eaten a healthy diet - also the very naive assumption that should 'healthy' foods be available, children would select them - where is the evidence for that in any child - it is the parents' choices and teaching that formulate childrens' behaviour in non Indigenous peoples so if that is not present in Indigenous peoples how are children supposed to know what is 'healthy' and what....moreover even if they do, what is the evidence that they would chose such foods?

The review itself is good and if that were presented as a review without all the other statements, it would be an acceptable paper.

Author Response

Thank you, please see the attached document outlining the reviewer's comments.
